# Designing Tangible as an Orchestration Tool for Collaborative Activities

Yanhong Li [1,*], Aditi Kothiyal [2], Thomas Weber [1], Beat Rossmy [1], Sven Mayer [1] and Heinrich Hussmann [1]

[1] Media Informatics, LMU Munich, 80337 Munich, Germany; thomas.weber@ifi.lmu.de (T.W.);
beat.rossmy@ifi.lmu.de (B.R.); info@sven-mayer.com (S.M.); hussmann@ifi.lmu.de (H.H.)
[2] Swiss Federal Institute of Technology Lausanne (EPFL), 1015 Lausanne, Switzerland; aditi.kothiyal@epfl.ch
*  Correspondence: yanhong.li@ifi.lmu.de

**Abstract:** Orchestrating collaborative learning activities is a challenge, even with the support of technology. Tangibles as orchestration tools represent an ambient and embodied approach to sharing information about the learning content and flow of the activity, thus facilitating both collaboration and its orchestration. Therefore, we propose tangibles as a solution to orchestrate productive collaborative learning. Concretely, this paper makes three contributions toward this end: First, we analyze the design space for tangibles as an orchestration tool to support collaborative learning and identify twelve essential dimensions. Second, we present five tangible tools for collaborative learning activities in face-to-face and online classrooms. Third, we present principles and challenges to designing tangibles for orchestrating collaborative learning based on our findings from the evaluation of ten educational experts who evaluated these tools using a usability scale and open questions. The key findings were: (1) they had good usability; (2) their main advantages are ease of use and support for collaborative learning; (3) their main disadvantages are limited functions and the difficulty to scale them to more users. We conclude by providing reflections and recommendations for the future design of tangibles for orchestration.

**Keywords:** tangible learning; orchestration tool; collaborative learning; human–computer interaction; tangible interaction; user interface; tangible user interface

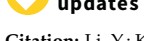



## 1. Introduction

Collaborative learning has many advantages, such as involving students actively in the learning process and improving learning outcomes [1]. However, successful collaborative learning requires specific interaction patterns to occur [2], and this often needs instructional intervention [3]. Previous work suggests three [4,5] to five [6,7] interactive patterns, such as positive interdependence and promotive interaction, for collaborative learning to be productive. To facilitate promotive interaction [7] and knowledge construction [8], teachers and instructional designers need to script collaborative activities and orchestrate them [2,9] to have desirable outcomes. Technology can facilitate this orchestration by providing teacher tools for monitoring a group's activity and intervening when necessary [2]. However, existing technologies (e.g., tablets and interactive whiteboards) have limited or preset interaction and communication patterns, which restrict the effectiveness of collaborative activities. This situation is because available technology devices do not have specific considerations for the requirements of collaborative work.

Tangible technology offers the potential to design orchestration tools for productive collaborative learning in complex, real-world learning situations [10,11]. This is because they offer the possibility of manipulating physical objects and exploring the physical world, which "facilitates both the acquisition of information about and experience with, the environment, together with an exploration of different combinations of information" [12]. Although previous studies have used tangibles to support collaborative learning [13–15], it remains unclear how to design such tangibles as orchestration tools for collaborative

learning [4,16]. We need a design framework that provides conceptual guidelines for designing tangibles as orchestration tools for collaborative activities. Specifically, the following questions emerge regarding the role of tangibles: *Whichtangible interactions can support effective collaborative learning beyond pure fun? How can we design such interactions? What information do teachers need to orchestrate productive collaborative learning? How can tangibles embed the interactive feedback information that teachers need for orchestration?*

In this paper, we analyze the design space of tangibles as an orchestration tool used for organizing and facilitating collaborative activities and identify twelve essential design dimensions. To demonstrate the use of this design space, we designed and developed five tangible tools for orchestrating collaborative learning activities, four (i.e., *stayFOCUSed*, *Group Hexagon*, *Tower*, and *Glowing Wand*) for face-to-face learning scenarios, and one (i.e., *Remolight*) for online classes. We invited ten educational experts to evaluate them. Unlike previous studies, we try to understand the role of tangibles in mediating collaborative interaction and communication from a design perspective. The expert evaluation indicated that our developed tangible tools had good usability, were easy to use, and supported productive collaborative learning interaction patterns. In addition, the experts found that the main disadvantages were limited functionalities and the difficulty of scaling them to more users.

In summary, the contribution of this paper is three-fold. First, we identified the design space of tangibles that support collaborative learning. Second, we share the experience of designing five tangible tools to orchestrate face-to-face and online collaborative learning. Finally, we propose principles and challenges to designing tangible tools for collaborative activities based on our findings from expert's evaluation. With this, we promote the design and use of tangibles as orchestration tools for collaborative learning in classrooms.

## 2. Related Work

To understand how tangibles could support the orchestration of collaborative learning, we examine the related work of tangible collaborative learning and tangibles as an orchestration tool to support collaborative activities.

### 2.1. Tangible Collaborative Learning

Collaborative learning refers to people working together in groups to understand a concept or solve a problem in order to benefit from the knowledge and skills of multiple people [2,5,17]. Physical affordance can change the meaning of an artifact and actions put on it, which enhance ownership, enable engagement, and facilitate awareness [18]. Studies from the fields of psychology [19], human–computer interaction [20], and technology-enhanced learning [21] suggest that collaborative learning with tangibles is an essential research area [11,22] because tangibles make the interactive interface intuitive and support collaborative activities better [20]. Currently, tangibles have been used in many collaborative scenarios, such as exploration [23–25], problem solving [26–28], skill development [29,30], and communication [15,31]. The main purposes of using tangibles [11] were to facilitate collaborative learning, teach skills such as programming and languages, and provide an embodied learning environment or experience.

Tangible collaborative learning has several advantages. First, interacting with tangibles promotes and enhances productive collaborative learning processes because tangibles provide access to shared representations of the problem, thus increasing the group working memory and reducing cognitive load [32,33]. Second, tangibles support collaborative activities by allowing multiple learners to simultaneously interact with the system [34], which could implicitly facilitate group communication and collaboration [35]. Third, tangibles have the advantage of creating flexible, collaborative learning environments [36], which can include whole-class activities and discussion [37]. Finally, tangibles can create interdependence, provide multiple perspectives to learners, and make learners aware of their peer actions and eye gaze, all of which promote productive collaborative learning processes [33].

*2.2. Tangible as an Orchestration Tool for Collaborative Activities*

Orchestration means to "manage (or subtly guide) the different activities occurring at different educational contexts and social levels, using different resources and tools in a synergic way" [38]. The orchestration process consists of managing class interactions at multiple levels of interaction: individual activities, teamwork, and class-wide discussions. Teachers use *orchestration tools* to capture, analyze, and visualize students' communication or progress in the learning activity [39–42]. Teachers could use such information to monitor or support students [43], which includes adapting activities in real-time according to student interaction and behavior. Technologies offer benefits for all parts of orchestration, which include planning, regulating, awareness, and intervening during collaborative learning [9,38].

We summarize three reasons why tangibles could help orchestrate collaborative learning activities compared to other technologies. First, a tangible acts as a physical object to embody learning knowledge. Thus, object manipulation becomes a process of knowledge internalization. It has at least three good examples. Rygh [44] found metaphors and affordances in physical objects were the reason why tangible tools support collaboration. *StoryBlocks* [45] was a tangible programming game where blind and visually impaired or sighted high school students create audio stories by combining code blocks, which helped novices learn computer science concepts. Sabuncuoglu [46] developed a tangible music platform wherein children could create a melody by placing the designed tangible blocks in an algorithmic structure. Baurley et al. [47] explored how tangible interfaces could capture and communicate embodied knowledge as a recipe authoring tool for innovative food, where users could use their bodies to learn ingredients. In addition, many studies used cubic shapes to include learners' behaviors [48]. By manipulating, placing, and arranging physical objects in space as the input, students understand abstract concepts easier and better.

Second, tangibles can act as interactive objects to embed or visualize individual and group activity-related information to support teacher awareness. Depending on the group's progress, the teacher may need to shift the students' attention to the learning requirements frequently. A tangible has the advantage of providing necessary information to orchestrate these changes. For example, *Lantern* [49] and *Shelve* [50] were designed to display teamwork information, such as which team is working on which exercise, how long they have been working on that exercise, whether they need help, and for how long. *FireFlies2* [51] was designed to convert the teacher's cognitive workload into distributed cognitive tasks, which helped the teacher focus more on adapting his or her instructions to students' abilities and needs. Ref. [52] designed tangible tabletops to help teachers manage the classroom and present visual information about students' progress. Finally, Baudisch et al. [53] developed *Lumino*, tangible blocks for tabletop computers, to demonstrate how to use tangible blocks to control a regular touch screen. The *Lumino* construction kit allows users to put together simple block constructions in which the system automatically checks the designs and problems of the hypothetical building.

Finally, the tangible can serve as a tool to facilitate communication and interaction, which aims at triggering specific types of collaborative learning processes that are known to generate learning gains, such as providing explanations or elaborations, resolving conflicts, or mutually regulating each other [54]. Compared to a tablet, the tangible *Quizbot* [55] made the children reach consensus easier and treat each other with more respect. *Paper-TUI* [56] used the social regulation approach to help users to create a web with digitally augmented physical papers, which helps identify and model interactions that support students' collaborative learning activity. *Sync Blocks* [57] coordinated children's collaboration by devising clear roles and reducing conflicts. In addition, Gelsomini et al. [58] explored a new Bring Your Own Device (BYOD)-based tangible technology-enhanced learning setup that supported the creation and management of storytelling activities and fostered the development of communication skills through mobile computer-supported collaborative

learning. Meanwhile, this approach could be extended to environments specifically crafted for individuals with special needs.

Based on the literature discussed above, we summarize that tangibles for orchestrating collaborative learning activities offer the opportunities of shared representations, simultaneous interactions, creating flexible learning environments, fostering interdependence, embodying knowledge, showing individual and group information and facilitating communication/interaction. Despite these opportunities of tangibles for orchestrating collaborative learning, it remains unclear how to specially design tangibles [4,16] for the orchestration of collaborative learning. What is the design space for tangible orchestration of collaborative learning? How to design tangibles as orchestration tools to support collaborative learning activities? We conjecture that tangibles can support collaborative learning, but we should design them appropriately to orchestrate productive collaborative learning processes. Therefore, we need a design space framework for tangible collaborative learning, which can help guide our understanding of how to design tangibles as orchestration tools to support collaborative learning activities.

### 3. Design Space for Tangible Collaborative Learning

*3.1. Design Requirements*

In order to clarify the design of tangibles for learning, Markova et al. [59] provided four criteria that a tangible must fulfill: (1) *Tangible Objects*: Contain one or more tangible objects as interactive devices; (2) *Embodiment*: Input and output are closely related temporally and spatially; (3) *Metaphor*: Digital and physical spaces are closely integrated; and (4) *Continuity*: Support continuous interactions. This list came from some earlier tangible characteristics proposed by Ullmer and Ishii [60]: (1) physical representations are computationally coupled to underlying digital information; (2) physical representations embody mechanisms for interactive control; (3) physical representations are perceptually coupled to actively mediated digital representations; (4) the physical state of interface artifacts partially embody the digital state of the system. These frameworks can help us avoid ambiguity about whether a system was a tangible interface or just a system with tangible aspects.

To design tangibles that are effective for orchestrating collaborative learning, we need to identify which factors have an impact on the effectiveness of orchestrating collaborative learning. Specifically, we need to ask: (1) *What information does the teacher need to support their students?* [43]; (2) *How to embed knowledge into tangibles to facilitate communication, such as teacher/teaching assistant–student and student–student communication?* [61], and (3) *How can tangibles support an orchestration process in a face-to-face or an online class, e.g., mathematical course?* [9,62,63].

*3.2. Design Framework*

As shown in Table 1, we developed a framework for the design space of tangibles as an orchestration tool, which has four basic elements we recommend to consider: user, context, collaboration, and interaction. From the HCI perspective, user, context, and interaction are essential considerations for tangible design [64,65]. From the perspective of orchestrating collaborative activities, tangible design needs to consider the aspects of the collaborative purpose, mechanism and scenario that need to be supported [66,67]. These elements emerge from integrating elements identified as essential for designing tangibles and orchestrating collaborative learning in previous work in education and HCI [43,67]. In addition, it summarizes the unique tangible designs for orchestrating collaboration. It is very beneficial for users with special characteristics, e.g., visually impaired. Interaction and communication also benefit from tangibilities, such as embodied input [11], physical representation [68], and interactive metaphors [69].

First, we need to know the users' type and number and analyze their characteristics. Children [70], teenagers [71], and adults [72] have different experiences and cognitive development. Group size affects the interactive system design since tangibles needed to

consider all the individuals contributing to the group work. Thus, the group size of more than six participants will make the tangible design more complicated than pairs or small groups (3–5 users). In addition, users' characteristics are a critical factor to influence the tangible design because these characteristics require us to make special considerations for how to design an interaction. For example, if the users are visually impaired, we need to design the interface in a recognized way (e.g., rough pattern) to know where and how to interact. At the same time, the interactive output should not be visual but provide audio and haptic feedback. Second, for the context, different modes of learning (i.e., face-to-face, remote, and blended) and locations (i.e., indoor and outdoor) need to be considered.

**Table 1.** Design space for tangible collaborative learning (unique tangible designs are indicated with green color ).

| Elements | Dimensions | | | |
|---|---|---|---|---|
| **User** | **Type** | Child | Teenager | Adult |
| | **Group size** | Pair (2) | Small group (3–5) | Large group (6+) |
| | **Characteristic** | Visually impaired (e.g., blind) | Action or perception impaired (e.g., stroke, impaired (autism, dyslexia) | Other general users |
| **Context** | **Mode** | Face-to-face | Remote | Blended |
| | **Location** | In-door (e.g., classroom, museum) | Outdoor (e.g., outing) | |
| **Collaboration** | **Purpose** | Problem-solving [73] | Brainstorming [74] | Knowledge building [75] |
| | **Mechanism** [7,66] | Interdependence | Coordination | Monitor learning process |
| | **Scenario** | Within the group | Between groups | |
| **Interaction** | **Input** [11] | Body-based gesture | Object manipulation | Move objects on interactive screens (e.g., tablet) |
| | **Physical representation** [68] | Symbolic | Literal | |
| | **Output** [68] | Visuospatial | Audial | Haptic |
| | **Interactive metaphor** [69] | Cartesian space | State space | Relational metaphors (human relations) |

Third, in order to support collaboration effectively, we need to understand its purpose, mechanism, and scenario. When reviewing the literature [73–75], we found three primary *purposes* for collaboration: problem-solving, brainstorming, and knowledge building. A tangible mainly acts as an orchestration tool during problem-solving and brainstorming. However, it acts as a knowledgeable object for knowledge building, where the movement or manipulation of tangible objects conceptualizes the internalization of new knowledge. The *mechanism* refers to the approaches taken to organize peoples' work together in order for the collaborative learning to be effective, and it has three common types [7,66]: interdependence, coordination, and monitoring learning processes. Positive interdependence links group members together. Each individual contributes to the success of the teamwork. In a collaborative learning environment, group members identify and build on each others' knowledge so that everyone contributes to the learning task. This process can take place only with coordination. In addition, learning is a highly interactive and dynamic process. For effective learning to occur in a collaborative learning environment, closely monitoring how students progress and collaborate in the process is crucial. Next, the *scenario* is a factor we need to consider because the communication and interaction within the group are more present, context-shared, and goal-centered than between groups.

Finally, considering interactions, *input and output interactions* are similar to the interactions of any tangible tools, with no special requirements for orchestration. However, tangible collaborative learning has two other important factors to be considered: *physical representation* [68] and *interactive metaphor* [69]. Physical representation encompasses design considerations of the representations themselves and how this corresponds to the artifact and action within the context or subject domain of use. Symbolic representation means objects that act as familiar signifiers, e.g., blocks, used to represent various entities, where the object may have little or no characteristics of the entity it represents. In contrast, literal representation refers to objects whose physical properties closely map with the domain's metaphor.

The design of *interactive metaphor* is related to collaborative purpose and mechanism. Learning settings have three forms: Cartesian space, state space, and relational metaphors or human relations [69]. In three-dimensional space, the Cartesian coordinate system comes from three mutually perpendicular coordinate axes: the *x*-axis, the *y*-axis, and the *z*-axis. Beyond graphical user interfaces, we can design tangibles to use the Cartesian space to help learners understand abstract concepts, e.g., geometry. A state space is the set of all possible configurations of a system. It is a valuable abstraction for reasoning about the behavior of a given system, and we can widely use it in the fields of artificial intelligence and game theory. The states of tangibles are changed directly by interactions, which makes users understand the concept immediately by the perceptions of the state changes. The fundamental metaphor when users describe tangible behavior in relational terms is to see emotional closeness as physical closeness. In a way, the metaphor works both ways in that physical closeness also creates emotional closeness. We are dealing with a *relational* universe with a topology different from the physical universe. Relations are convenient to be shown and understood by the tangible physical representation, often designed with a meaning map.

### 3.3. Design Rationale

Our underlying assumption was that this works' emphasis on collaborative spaces and interactive methods made a difference between tangible and traditional interfaces. However, such interfaces have to fit well with the pedagogical concepts of a learning experience. Cross-plane integration, sequentiality, time management, and physicality are essential considerations for an effective collaborative learning experience [54]. Putting these requirements together, we designed tangible tools that create a shared space for communication, enable within-group and inter-group communication and help-seeking, and facilitate activity status sharing (e.g., answers, remaining activity time). In the following, we report on five prototypical tangible tools (see Section 4), which cover different regions of the design space. We developed these tangible tools for orchestration in an iterative process involving students and potential tool users. We discuss the development process and the detailed functionality of the prototypes in Section 4.

#### 3.3.1. Creating Shared Spaces for Communication

As an orchestration tool, the tangible needs to enable shared spaces for communication, where the collaboration can happen. Tangibility involves gesture, motion, or full-body interaction and "emphasizes the use of the body in educational practice" [76]. By embedding technology in physical objects with natural actions such as grabbing, tangible becomes ubiquitous, mixing the physical and digital world [77]. As shown in Figure 1, we developed five prototypes that had different communicative mechanisms. *stayFOCUSed* acts as a tool to create a collaborative atmosphere, where students have to get close and finish the activity together. The behavior to cast light on the ceiling is an excellent method to gather students and attract their attention. *Group Hexagon* gives each student an individual device to interact with; then, it connects these with a central group device. This process helps to build among students a sense of individual-community connection. *Tower* has both a group device and an app to make students share their opinions. The *Glowing Wand*

is a playful and fun device for each student. It has no direct affordance for group work; however, this embodied and present behavior naturally attracts students to work together. *Remolight* connects individuals at different locations, where they have the same device as an ambient environment to convey important information.

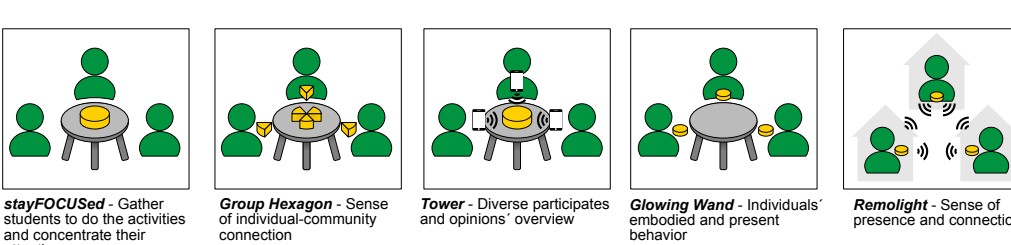

**Figure 1.** Communicative mechanisms for developing tangibles in our paper.

### 3.3.2. Supporting Diverse Interactive Dynamics

An orchestration tool needs to support different types of communications (e.g., within-group and inter-group) and help-seeking, a primary function for group discussion with supervision from teaching assistants or teachers. As shown in Table 2, we designed different communicative approaches within the prototypes. All the prototypes can realize the communication and interaction within the group in different ways. For example, students write down their answers or questions, and *stayFOCUSed* will project them to the classroom ceiling. *Group Hexagon* has designed an individual hexagon for all the students in the group. Students touch side senses to turn on/off the light to communicate with peers in the group. *Tower* has a magnet tablet design with white and green colors. Students can pick up a white or green one to paste on the different levels of the tower. The *Glowing Wand* can be moved in different gestures to change its color. If the students in the group have a common understanding of the meaning of different colors, they can effectively communicate. Students can squeeze the ball of *Remolight* to show an agreement for others' opinions in the group.

Inter-group functions are helpful for students to communicate and interact beyond the group. Because students in different groups often do not sit together, it requires a simple and clear tangible design. As shown in Table 2, *stayFOCUSed* and *Glowing Wand* keep the within-group design for inter-group interaction. However, *Group Hexagon* designs an additional tangible object to show group work, which is a "group hexagon". The group hexagon changes to a green or red color to show the status of group work. *Tower* provides an app to enable communication via online learning. For our tangible tools, we assume a setting that distinguishes between different roles of advisers: teacher and teaching assistants [62,78]. The difference in communication and interaction between the teacher and teaching assistants is that we assume there is only one teacher for many groups, and he/she is not physically present in all the classrooms. However, we assume teaching assistants are always in the classroom with the supervised group. Thus, only *Group Hexagon* and *Tower* have designed an app to interact with the teacher, such as managing the activity (e.g., start the activity) and knowing the discussion in each group. The use context of *Remolight* is online and remote. Thus, it is the same interaction to squeeze the ball to send a light animation to the teacher or teaching assistants.

The purpose of interacting with teaching assistants is for help-seeking. In a real class, there are several groups. Thus, the signal should be eye-catching. *stayFOCUSed* uses the ceiling projection for this. *Tower* makes the top bulb flash with a red color to catch the teaching assistant's attention. The *Glowing Wand* turns into rainbow colors. Group objects in the *Group Hexagon* would flash to show their help-seeking. The communication between different classrooms mainly relies on the app. Only *Group Hexagon* and *Tower* include this kind of consideration. It appears as a learning forum to exchange students' problems and ideas.

**Table 2.** Communication and interaction (**C and I**) designs in our paper (**G**: Group; **TA**: Teaching Assistant; **T**: Teacher; **CR**: Classroom).

| C & I | *stayFOCUSed* | *Group Hexagon* | *Tower* | *Glowing Wand* | *Remolight* |
|---|---|---|---|---|---|
| within **G** | Overhead Projection | Individual-Hexagon | Magnet Object | LEDs | Ball |
| inter-**G** | Overhead Projection | Group-Hexagon | Top Bulb, App | LEDs | - |
| with **TA**s | Overhead Projection | Group-Hexagon | Top Bulb | LEDs | Ball |
| with **T** | - | App | App | - | Ball |
| within **CR** | Overhead Projection | Group-Hexagon | Top Bulb, App | LEDs | Ball |
| between **CR**s | - | App | App | - | - |

### 3.3.3. Visualizing Interaction and Activity Status

Orchestration is similar to a regulation loop, which has two concrete points of control: state awareness and workflow manipulation [79]. The notion of "awareness tools" [80] is to inform users about the activity of their co-workers, where awareness shares behavioral information among users without a cognitive diagnosis. In order to provide dynamics for consistent group communication, we stress the need for interactive information visualization in the design of orchestration technologies. This paper designed light (brightness and color) to show the interaction and activity state information. *Minimalism* in the design of orchestration technologies with light was emphasized by Dillenbourg et al. [79] and has been used in their design of *Lantern* [49]. Based on that, we explored more possibilities to design information visualization with light, e.g., overhead projecting to the ceiling (*stayFOCUSed*) and hexagons with different colors (*Group Hexagon*).

In order to maintain the collaborative activity, it is essential to provide and visualize some basic interactive information, e.g., activity time and help request. Therefore, we designed the prototypes to ensure that they support the visualizations of different interactive information (see Table 3). Furthermore, using an app has shown to be a good way for open discussion and help-seeking. However, it has no design of tangible interaction and therefore is somehow outside the core scope of our work.

**Table 3.** Collaborative learning activities supported in our paper (**MCQA**: Multiple Choice Question Answer; **OQA**: Open Question Answer; **HR**: Help Request; **TA**: Teaching Assistant).

| Activities | *stayFOCUSed* | *Group Hexagon* | *Tower* | *Glowing Wand* | *Remolight* |
|---|---|---|---|---|---|
| Submit **MCQA** | Write on disks | Turn on individual hexagon | Attach magnet object | Move wand | Change the ball color |
| Submit **OQA** | Write on disks | App | App | - | - |
| Set activity time | Progress Light Bar (**PLB**) | **PLB** | **PLB** | - | **PLB** |
| Share **MCQA** | Overhead projection (**OP**) | Connect group hexagon | Read magnet | - | - |
| Share **OQA** | **OP** | App | App | - | - |
| Finish activity | **OP** | Green light (**GL**) | **GL** | **GL** | - |
| **HR** for TAs | **OP** | Light flashing | Top bulb flashing | Rainbow light | Light flashing |
| **HR** for remote teacher | - | App | App | - | - |
| **HR** for near groups | **OP** | App | App | - | - |
| **HR** for remote groups | - | App | App | - | - |

## 4. Tangible Development

Our aims to design tangibles for orchestrating collaborative learning activities were: (1) Support collaborative activities in the class (face-to-face and remote); (2) explore what information to show; (3) how to display information to facilitate collaborative communication.

During two university computer science courses in 2020 and 2021, twenty-three computer science master students supervised by a team of four HCI researchers developed eight tangible prototypes for orchestrating collaborative learning in the face-to-face classroom and online lecture. The motivation to explore the idea of tangible orchestration in the computer science practical courses was to explore more possibilities, such as conceptual ideas, and observe the actual products. As shown in Table 4, our developed prototypes were designed for typical university students working in small groups, which are either in face-to-face or online classes. The primary purpose of the collaboration is problem-solving, and the orchestration tool is used for monitoring learning processes within and between the groups.

**Table 4.** Design dimensions we used in this paper are indicated with peach background color .

| Elements | Dimensions | | | |
|---|---|---|---|---|
| **User** | **Type** | Child | Teenager | Adult |
| | **Group size** | Pair (2) | Small group (3–5) | Large group (6+) |
| | **Characteristic** | Visually impaired (e.g., blind) | Action or perception impaired (e.g., stroke, autism, dyslexia) | Other general users |
| **Context** | **Mode** | Face-to-face | Remote | Blended |
| | **Location** | In-door (e.g., classroom , museum) | Out-door (e.g., outing) | |
| **Collaboration** | **Purpose** | Problem-solving [73] | Brainstorming [74] | Knowledge building [75] |
| | **Mechanism** [7,66] | Interdependence | Coordination | Monitor the learning process |
| | **Scenario** | Within the group | Between groups | |
| **Interaction** | **Input** | Body-based gesture | Object manipulation | Move objects on interactive screens (e.g., tablet) |
| | **Physical representation** [68] | Symbolic | Literal | |
| | **Output** [68] | Visuospatial | Audial | Haptic |
| | **Interactive metaphor** [69] | Cartesian space | State space | Relational metaphors (human relations) |

We provided a university mathematical class as the user context. More specifically, two activity scenarios [78] were given: **peer instruction** and **community-supported worksheets**. Peer instruction strategy is a "clicker question" (single/multiple choice question) strategy and consists of seven steps [81]: (1) pose question; (2) students work on the problem; (3) students submit their answers; (4) discuss in the group (peer discussion); (5) submit revised answers; (6) giving feedback to the teacher, and (7) teaching assistant explaining the solution. This learning activity is suitable for pairs or small groups (3–5 students). The community-supported worksheet strategy includes two worksheets (e.g., integral problems). The first one is easy to solve, but the second one is more complicated and challenges students' existing knowledge. Students who complete the first part must add "hints" for others and be encouraged to solve the second part. Students who have not completed the second part can use the hints and then "upvote" the most helpful ones. When designing and developing tangibles for orchestration, all prototypes should focus on supporting group processes in the assigned scenarios (i.e., peer instruction and community-supported worksheet). As shown in Figure 2, it should consider the communication and interaction within the classroom, across classrooms, with the teacher and with teaching assistants.

**Figure 2.** User context of developed orchestration tools in our paper.

Each prototype development took a whole semester (around four months) and followed an iterative design process to generate insights based on the research through the design approach [82]. As milestones for the iterative process, we used: (1) concepts presented as storyboards [83], (2) paper prototypes [84], and (3) experience prototypes [85] to constantly re-evaluate the designs and integrate insights from experiments, discussion and previous design iterations. Final prototypes (four for face-to-face and one for online collaborative learning) have practical physical functions, including the required casing, sensors, actuators, and electronics.

### 4.1. stayFOCUSed

*stayFOCUSed* is a tangible tool that uses light projection on the ceiling and light feedback on the device to support collaborative learning activities (see Figure 3-3). Light feedback on the device supports group work, which indicates the remaining time via a progress bar in traffic light colors (see Figure 3-2). During exercise tasks, students are supposed to use small disks to choose multiple-choice-task answers (see Figure 3-4), and they need to place it over a light beam. To uncover the result of voting, we can rotate the projector lens to focus the light beam (see Figure 3-3). Subsequently, students can discuss the outcome of the poll. We can use different colored disks to send group work status (green = finished, red = help) to other groups and the teaching assistant (see Figure 3-1). Empty disks can even be used to write and share information by freehand writing (see Figure 3-4). This hand-writing allows us using *stayFOCUSed* in a new learning activity or different contexts. The projection on the ceiling further uses the students' peripheral perception to place information outside of the usual working context to prevent information overflow and generate ambient feedback for shared information. We have three main parts of technical designs: (1) Progress bar has 5 LED strips with 6 LEDs on the lower part of the *stayFOCUSed*. In order to project the light, we implement 12 super bright LEDs on a mini breadboard inside of a tube at the bottom. We coded them with an Arduino Uno; (2) we used Fresnel lens and Collimator lens for the "focus effect"; (3) we printed the whole lamp with transparent Polyethylene terephthalate glycol (PETG).

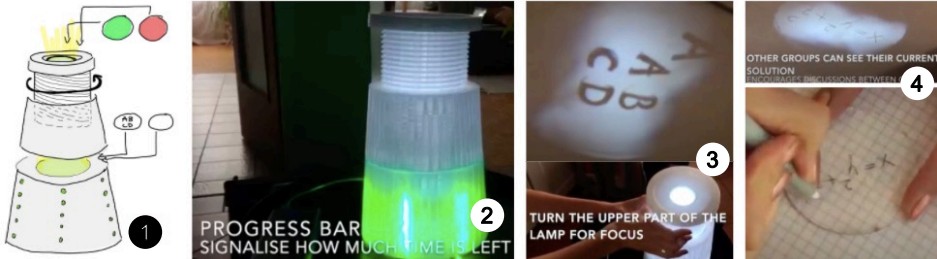

**Figure 3.** *stayFOCUSed* technical prototype (**1**. concept idea and structure; **2**. show timer with light progress bar; **3**. rotate the lamp to show answers; **4**. pen-writing on the disk).

## 4.2. Group Hexagon

*Group Hexagon* is a modular tangible that supports different group learning activities. Through a secondary smart device (see Figure 4-5), the teacher can change the operational mode of *Group Hexagon*. Each group has one group hexagon and six individual hexagons (see Figure 4-5). Individual hexagons can connect with the group hexagon, for instance, to show the summary information of the learning activity. In addition, the individual hexagon can display the remaining time of the learning activity (see Figure 4-2). The group hexagon shows the answer distribution for multiple-choice tasks. We can use the group hexagon to send and show the group operational status to the other groups and the teaching assistants (see Figure 4-4). The individual hexagons are used in detached mode by the students to pick answer options (see Figure 4-1). If connected to the group hexagon, they may show solutions of working tasks (see Figure 4-5). When interacting with *Group Hexagon*, we can use touch gestures to select a task and do miscellaneous interactions. Colors show different information. The group hexagon functions as a low display that can show quantity information using light visualizations (see Figure 4-4). There is a Bluetooth connection between group-individual hexagons and with the smartphone. There are six copper touch-buttons on each side of the individual hexagon and one on the top of the group hexagon. Inside each hexagon there are LED strips, an Arduino Uno, cables, a Bluetooth module, and a power bank.

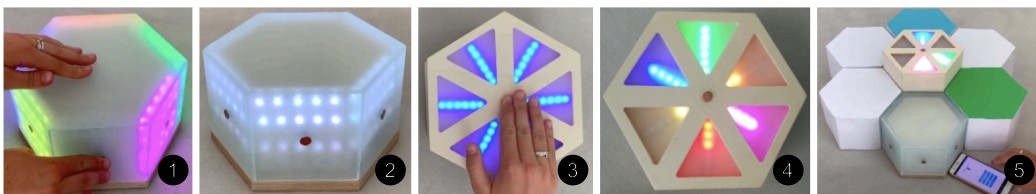

**Figure 4.** *Group Hexagon* technical prototype (**1**. touch side button to choose an answer with individual-hexagon (**IH**); **2**. show timer with light progress bar in the **IH**; **3**. touch the top button to seek help with the group-hexagon (**GH**); **4**. show answer distribution in the **GH**; **5**. app mode to interact and control **IH**).

## 4.3. Tower

We designed *Tower* to facilitate the interactions within and across groups. By placing magnets on the outer grid on the device surface, students can participate in voting on multiple-choice questions (see Figure 5-2). We use different colored magnets to indicate students' certainty regarding their answers. The grid rows demonstrate the response options, and the columns represent the individual group member's workspace. The color-coding of the rows adapts depending on how many members voted for each answer option highlighting the answer distribution and stimulating a discussion about the correct answer options (see Figure 5-2). In addition, we can use the top bulb of the *Tower* to seek help and signalize the operational status (see Figure 5-3). For communication and interaction with other groups, students can use a mobile app (see Figure 5-1). These interactions are (1) call for help from peers, (2) provide help to other groups, (3) rate other groups' work, (4) rate your activity progress, or (5) participate in discussions. Actual physical materials such as paper can be used with the magnetic interface surface to adapt to individualized processes and upcoming learning activities. We implemented a touch function with the copper foil. If the user touches the bulb, white LED pins turn on; if it touches again, white LED pins turn off. Magnets stick on the *Tower* because it has a metal strip inside. The reed switch recognizes the magnet and passes on the signal to the LED stripe. The LED stripe lights up in the respective color.

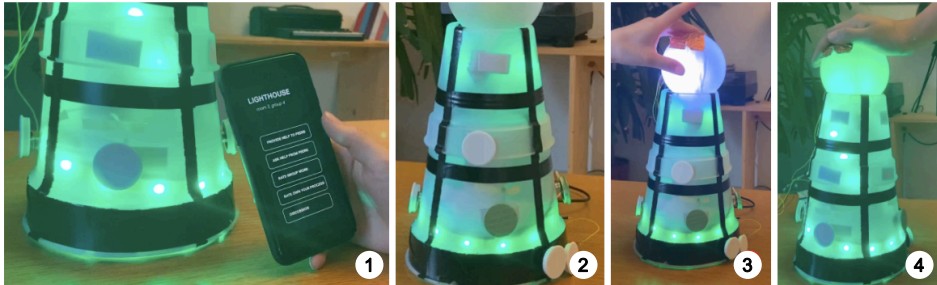

**Figure 5.** *Tower* technical prototype (**1**. use app to communicate with other groups; **2**. place magnets on the *Tower* to choose an answer, green means *I am confident*, white means *I am not sure*; **3**. rotate top bulb to seek help; **4**. touch top bulb to finish the activity).

### 4.4. Glowing Wand

*Glowing Wand* is a personal handheld tangible that students use to participate in the learning activities. The idea comes from a magic wand, and thus, we use motion gestures to control the *Glowing Wand* (see Figure 6-1). Different gestures indicate to change the *Glowing Wand*'s color, whereas the inclination regulates its brightness. The combination of color and brightness communicates the student's current operational state or quickly overviews the participants' opinions in voting situations. We consider simple gestures broadly understandable and associated consistently with the traffic light color schemes. For example, a tick gesture picks the green light (see Figure 6-5), a circle gesture picks yellow (see Figure 6-4), and a negative tick changes the color to red (see Figure 6-3). This system fits well into learning activities, but we can also use it in self-defined cases or open group processes such as voting due to its open design and tool character. We implemented all the hardware at the bottom of the *Glowing Wand*, which contained Arduino, Gyroscope, Accelerometer, power boost (3.7 V), battery (5 V), and a charger. We repeated the gestures and obtained the Gyroscope log data, and later we used the ML algorithm to create the model for detecting gestures to change the LED colors in the *Glowing Wand*.

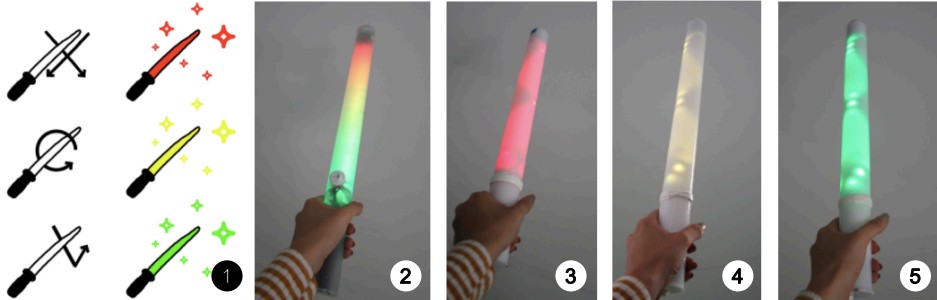

**Figure 6.** *Glowing Wand* technical prototype (**1**. different hand gesture designs; **2**. switch for a rainbow feedback; **3**. negative tick gesture to red light; **4**. circle gesture to yellow light; **5**. tick gesture to green light).

### 4.5. Remolight

Unlike the previous four prototypes, *Remolight* is a tangible device designed for remote collaborative learning. We can use it in the synchronous online group tutorial, where each student has a *Remolight* close to his or her learning device (e.g., computer, laptop, and tablet). It aimed to reduce technology distraction while learning, only when help-seeking or important notifications attract students' attention. It keeps basic functions, e.g., timer (see Figure 7-1), notification (see Figure 7-2), and help-seeking (see Figure 7-3). Users can use the squeeze interaction with the ball to connect with others, such as sending an agreement. We implemented a pressure sensor in the ball and realized functions of shaking and rotating to change the color with the 3-axis Gyroscope (LSM6DS33). For the squeeze function, the ball had a magnet on one end and a sensor on the other side. When the distance between the

sensor and the magnet was reduced due to squeezing, the magnetic value measured by the sensor increased. The color of the RGB LED changed from green to yellow to red as the magnetic force value changed from small to large.

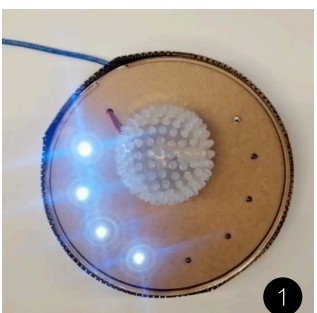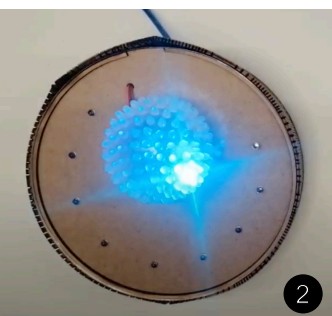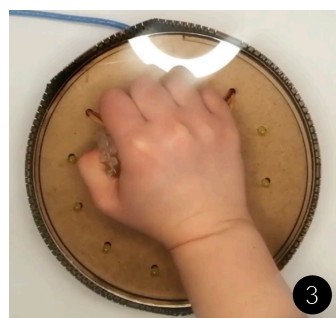

**Figure 7.** *Remolight* technical prototype (**1**. indicated timer with LED bar; **2**. show notifications with light shining in the ball; **3**. squeeze the ball to seek help or knock it to send an agreement message to other learners).

## 5. Expert Evaluation

We conducted an expert evaluation to understand the feasibility and practical value of the developed tangible tools for classroom orchestration. We opted for educational expert interviews over a user study as experts are more aware of orchestrating collaborative learning with tangibles because it was necessary for this investigation that they all understood the user contexts (i.e., peer instruction and community-supported worksheet scenarios).

### 5.1. Procedure

We conducted the evaluation using a video conference meeting. The whole process lasted for around one hour and had three main steps. First, we explained the design concepts, the purpose of the prototypes, and the questionnaire to the participants. Second, we introduced *Tower*, *Group Hexagon*, *stayFOCUSed*, *Glowing Wand*, and *Remolight* with slides, which contained high-quality videos to show the functionalities and interactions. Third, we answered the experts' questions about things that were unclear to them and then sent them an online questionnaire. Regarding how video evaluation is feasible for tangible devices compared to actual-world evaluation, we had several methods: (1) For each prototype, we provided a prototype function video and a user experience video to help participants know the actual effects. (2) Before showing the videos, we introduced each prototype's concept idea, from the sketching and paper prototype to the hardware details. (3) We provided the participants the chance to ask further questions before filling out the questionnaires.

The questionnaire included seven items, which we adapted from Laugwitz et al. [86]'s user experience questionnaire, which initially came from the AttrakDiff [87] questionnaire. We only chose seven items because some are irrelevant for our online expert assessment. We evaluated each item from 1 (e.g., *not understandable*) to 5 (e.g., *understandable*). In addition, we asked three open questions: *What do you think are the advantages of these tangible prototypes? What do you think are the disadvantages of these tangible prototypes? What are your suggestions to improve these tangible prototypes?*

### 5.2. Participants

We invited ten educational experts, one educational professor, and nine educational Ph.D. candidates to evaluate all of our developed tangible prototypes online. Six of the experts identified as female and four as male. Their average age was 35.6 (min = 28, max = 59).

### 5.3. Use Experience

Overall, all five prototypes ranked high on the seven items of our evaluation, see Figure 8. As we can see, except *Remolight*, only five ratings are under 4, which are *Group Hexagon*'s understandability (3.5 [2, 5]) and easiness (2.5 [2, 5]), *Glowing Wand*'s practicability (2.5 [2, 4]) and *stayFOCUSed*'s expectation (3.5 [3, 4]). There were four interesting findings of high score and consistency: (1) *stayFOCUSed* and *Glowing Wand* have a high consistency of understandability; (2) *Tower* has a high agreement about easiness, innovation, and practicability; (3) *Group Hexagon* has an undisputed attractability; (4) *Remolight* has differentiation in all aspects. In addition, when comparing prototypes, we can see: (1) *Group Hexagon* is outstanding for innovation, interest, and attractability, but hardest to understand and most complicated; (2) *Glowing Wand* is the most easy and understandable, but least practical; (3) *Tower* has high scores (equal or above 4) in all items, and *stayFOCUSed* has average ratings.

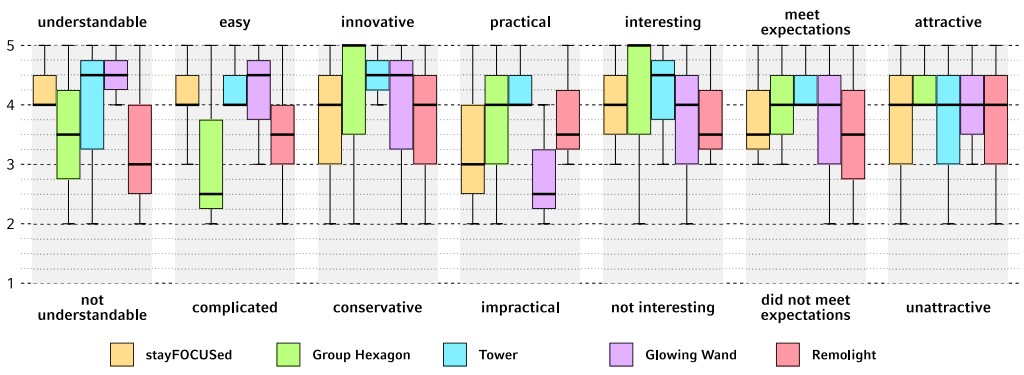

**Figure 8.** Median and range of developed tangible tools' usability in our paper.

### 5.4. Qualitative Feedback and Perspectives

In order to obtain detailed usability perspectives from educational experts, we asked three open questions. After introducing each of our developed tangibles, they wrote down their answers individually. Therefore, the evaluations we obtained from them are independent and not influenced by each other. This information helps us have a general understanding of how these tangibles can be more suitable in the actual classroom.

We coded the common themes among the expert responses. As shown in Table 5, we identified the following advantages of the developed tangibles. First, "easy to use" was the most common comment, which appears seven times for *Tower* and three times for *Glowing Wand*. Second, a similar comment was found for "phone App", it showed in *Group Hexagon* and *Tower* twice each. Third, the idea of "potentials for different application" had been mentioned in *stayFOCUSed*, *Group Hexagon*, and *Glowing Wand* with the statements such as "many potential purposes", "many possibilities", "many possible related ideas". Fourth, experts proposed *interesting* almost everywhere, e.g., "interesting and attractive" in the *Group Hexagon*, "interesting and fun for game-based learning" in the *Tower* and "low tech but fascinating solution", and "brings out the childhood memory" in the *Glowing Wand*. Finally, the experts mentioned the potential for collaborative learning, group interaction, and class management concerning all the prototypes. In summary, some keywords used concerning the prototypes included: fun, collaborative learning, and class management. From an educational expert perspective, these might constitute the value of using tangibles for collaborative learning.

**Table 5.** Educational experts' feedback on the advantages of this study's developed tangible tools (* number of comments obtained).

|  | **Advantages** |
| --- | --- |
| *stayFOCUSed* | - engage wider community participants (3 *)<br>- very flexible design for many potential purposes (2)<br>- multiple representations (2)<br>- non-verbal communications through lights (2)<br>- good for managing large classrooms (2) |
| *Group Hexagon* | - good for collaborative learning (5)<br>- interesting and attractive (3)<br>- many possibilities for other learning scenarios (3)<br>- phone app (2) |
| *Tower* | - easy to use (7)<br>- interesting and fun for game-based learning (2)<br>- phone app (2)<br>- interaction within and across groups (1)<br>- good for elementary student (1) |
| *Glowing Wand* | - low tech but highly interesting solution (5)<br>- easy to use (3)<br>- many possible related ideas (2)<br>- brings out the childhood memory for people who are participating or watching (1)<br>- can become a part of the expected grammar of interaction in the classroom (1) |
| *Remolight* | - easy to use (5)<br>- graspable gesture is interesting (4)<br>- simple, but satisfies the purpose (2)<br>- connect people (2) |

Walking to the other endpoint, our understanding will be refreshed and become more rational upon seeing the disadvantages' description (see Table 6). Unlike the advantages, educational experts' attitudes toward drawbacks looked diverse. However, if we generalize and summarize these viewpoints, three clusters are obvious. First, experts mentioned very straightforwardly that actual interaction in the classroom might be confusing and difficult. For example, experts mentioned "may not be easy for students to remember various rules of how to use the light", "students might be confused during activities", "students need to be trained for using", "difficult to see how this would work in a practical classroom", and "complicated for young students." Second, prototype design problems existed. In particular, we had three dilemmas: (1) tangibles have limited functions, which are consistent with the highly valued advantage of "possibilities." For example, experts expressed some issues such as "suitable for short group answers, but not a long one", "no consideration of colorblind people", "functions are too specific", and "difficult to read the answer that is projected on the ceiling"; (2) design had no irreplaceable features, which was conveyed by "compared with the computer screen, the advantages are not so obvious", and "it is replaceable by many other existing objects"; (3) prototype was not realistic: "prototype is too bulky to be practical." However, it was normal to have a bigger size to configure electronic components for initial HCI prototype development. Finally, the reasons for interaction might not be profound, as shown by the remarks "interactive process are redundant and not helpful", "might become distracted or unsafe if many students swing their wand." These comments took us back to some classic questions: *How do people learn? What tangibles are helpful for collaborative learning, not just to have fun? How do you know the actual effects of tangibles?* In summary, when looking at the disadvantage picture of tangibles for collaborative learning, the center of clusters were related to the utility of the *interaction process*, *function designs*, and *interaction purpose*.

**Table 6.** Educational experts' feedback on disadvantages of this study's developed tangibles (* number of comments obtained).

| | Disadvantages |
|---|---|
| *stayFOCUSed* | - difficult to read answers that are projected on the ceiling (2 *)<br>- may not be easy for students to remember various rules of how to use the light (1)<br>- suitable for short group answers, but not long one (1)<br>- limited interactions with the whole class (1)<br>- compared with computer screen, the advantages are not so obvious (1) |
| *Group Hexagon* | - difficult for the teacher and teaching assistants to manage the class (3)<br>- students might be confused during activities (2)<br>- complicated for young students (2)<br>- no consideration of colorblind people (1)<br>- students need to be trained for using (1)<br>- difficult to see how this would work in a practical classroom (1) |
| *Tower* | - interactive processes are redundant and not helpful (4)<br>- prototype is rough (1)<br>- functions are too specific (1)<br>- voting would have a negative impact on the minority students (1) |
| *Glowing Wand* | - usages should be very explicit, so not just a toy, but a real tool supporting students' individual or group work (2)<br>- it is replaceable by many other existing objects (1)<br>- prototype is too bulky to be practical (1)<br>- might become distracted or unsafe if many students swinging their wands (1)<br>- interaction is interesting but not meaningful (1) |
| *Remolight* | - functions are limited (3)<br>- might confuse students when they have to remember different colors (2)<br>- it can be easily replaced by smart phone (2)<br>- can be expensive to give each student such a prototype (2) |

In order to improve the prototypes, we also invited experts to provide improvement suggestions. In general, most of the suggestions reflected similar advantages and disadvantages. However, there were three inspirations, which might bring innovations for tangibles. First, "contain the entirety of the students' study life". It implied the integration of tangible learning, quantified learning, and customized learning. Customized and quantified learning refer to the need to know students' study behavior and preferences to recommend an optimal learning plan. Therefore, tangibles could be a suitable embodied object that accompanies students throughout their learning journey. Second, "be intelligent and social". We were not surprised about *intelligent*, but *social* refers to how to help students have a more engaging learning experience. In other words, tangible interaction already involves learners' embodied engagement, and together with "social engagement", they might have an enhanced learning experience. Finally, "use unambiguous instructions". This echoes with the sound of "actual interaction in the classroom might be confusing and difficult" from the disadvantages. Once again, we understood the importance of having both students and teachers know how tangibles help their activities.

## 6. Discussion

In this section, based on the findings from the expert evaluation, we discuss the importance of designing tangible interactions to increase personal connectedness and shared attention. Then, we reflect on and summarize the challenges and principles for designing tangibles for orchestrating collaborative activities. Finally, we recommend a new approach to creative design and reflect on this paper's limitations.

### 6.1. Tangible Interaction to Increase Connectedness and Shared Attention

For an effective collaborative activity, we should design the interaction with tangibles to (1) support the requirements of the activities, e.g., submit the answer to multiple-choice questions in the peer instruction scenario, (2) increase learners' sense of connectedness and shared attention. In our paper, tangibles have been designed to gather students to do their activities, concentrate their attention, increase their sense of individual-community connections, and facilitate individuals' embodied behaviors. Interactions with these tangibles were playful and exciting. Beyond it, they were also conceptualized and built with the individual-community associations. In other words, an individual's interaction connects or contributes to group work, which gives meaning to working together. In order to design tangible interactions that can support effective collaborative learning, except for an iterated design with full consideration of interactive function, meaning, and experience, we might also need to consider teachers' requirements and use context.

Our work shows that tangibles could embed interactive information, such as timing, answer, help-seeking, and activity status. All this information is considered necessary for group activities in the classroom. We see that some required information needs to be shared when considering orchestration. For example, group status (e.g., activity time, group progress, and individual's contribution) should arouse group members' and teachers/teaching-assistants' shared attention. Tangibles should make the interaction more straightforward and related to the group status. Meanwhile, it makes group information easier to perceive and understandable. Therefore, we need to consider the interactive feedback as an input or reason to forward the group work. In other words, the tangible should convey the information that could promote the group process.

### 6.2. Challenges to Designing Tangibles for Collaborative Activities

Coordinating and monitoring the learning process is a necessity and a challenge for the design of orchestration tools. We have two challenges in designing an appropriate tangible for collaborative activities. First, tangible design is activity-determined and should be available for other possibilities. As we know, collaboration and learning only occur in a concrete learning scenario. It is easy to design interesting tangibles without defining a learning activity, but their interaction does not support learning communication. Our study defined the tangible use context as a university mathematical class, clarifying the requirements. However, it was still challenging to satisfy diverse interactive requirements because tangible design should be flexible to give learners more freedom to communicate. For example, for *Group Hexagon*, the learner can interact directly with his or her hexagon but also can use a smartphone to do it. In addition, we were always careful not to restrict use possibilities because it was impractical to have teachers use a tangible only in a specific activity.

Second, tangibles' actual usability and affordance in a classroom are hard to predict. Education is an area that heavily cares about the usability and affordance of learning tools. Without seeing obvious benefits, teachers or students are not motivated to use it in practice. If we only consider the design elements of tangibles, it has several unique characteristics such as playful, fun, and engaging. However, putting it in a practical learning context, we may not see the expected effects. For example, *stayFOCUSed* had the advantage of using a shared space (i.e., ceiling) to show and display information. However, if the ceiling is too high or the classroom is too bright, it may not work. This situation poses a dilemma to design tangibles: some people are more likely to see the possibilities and contribution of new thoughts, which is beneficial to cultivating a new research area. Others might question and criticize the practical usage and effects, which is advantageous for producing high-quality studies. However, this study brought to light the difficulties of using tangibles in an actual class, which requires having a prototype with high-fidelity and technical stability and being used for enough time to see the effects.

### 6.3. Making Tangibles Flexible, Simple, and Fun

The tangible design includes considerations of collaborative interaction and learning but might overlook the feasibility of practical classroom implementation. Being flexible, simple, and fun does not sound like a significant discovery of tangible design principles for orchestrating collaborative activities. However, if HCI researchers want to design tangibles that can be used and be helpful in the actual classroom, it is essential that they have flexibility, simplicity, and are fun to use. A classroom is a study place for different topics and subjects. Correspondingly, the teacher designs group learning activities differently. We can design tangibles for a specific purpose, severely hurting their actual use in the classroom since we will be able to use tangibles for one specific topic in one specific class, thus creating a significant technical overhead not commensurate to its effect on the learning experience. In this situation, *flexibility* is a very high value because it provides more possibilities for teachers to design their expected learning activities using a generic tangible. In order to create an excellent collaborative learning experience in the classroom, tangible design needs to consider many situations and aspects, e.g., time, safety and effectiveness. Thus, a *simple* tangible can work better because it is more understandable and may offer a broad range of interpretations. Further, it is easier to use since teachers do not need to spend too much time explaining the instruction. We always design tangibles to have a new HCI experience, fun, exciting, and attractive. *Fun* is positive feedback that our brain gives us for learning. It is related to learners' motivation and flow [88], which are two critical factors that affect student learning.

The trap of tangible learning is that fun might not improve learning. There are four types of fun: easy, hard, serious, and social [89]. Easy fun means people's feelings of surprise, curiosity, and wonder, where the user context designs elements for exploration, creativity, and fantasy. However, serious fun makes people enjoy the interactive experience, where users produce emotions to promote mental activities. Social fun comes from user interactions, e.g., communication and cooperation, which is closely related to making tangibles for group learning more effective. To have tangibles benefit from serious fun and social fun theory, we need further cooperative studies from multidisciplinary areas, e.g., HCI, education, and psychology. Because it is a Jigsaw puzzle problem or dolomite mirror, people from a specific subject can only see the problems that use their domain knowledge. We could observe the effect of taking a viewpoint from another discipline in our study. Examples are comments from educational experts: "interaction is interesting but not meaningful", and "usage should be very explicit, so not just a toy, but a real tool supporting students' individual or group work". Therefore, here we provide a good direction to solve the problem more generally, using our proposed design space framework to design and develop tangible prototypes and conduct more studies about how these improve the serious and social fun of learning with tangibles.

### 6.4. Reflection and Recommendation for Creative Design

In addition to obtaining some insightful perspectives from educational experts' evaluations, we think it is valuable to share some information about finding innovative and creative ideas for tangibles for orchestrating collaborative learning. We supervised 23 HCI master students to design and develop tangibles for collaborative activities in this project. It turned out that the range of ideas developed showed high creativity and good technical feasibility. Our paper demonstrated two approaches that the teaching team followed. First, except scaffolding the acquisition of learning content and technical skills, we were able to help students become aware of their development process to take on more responsibility and ownership of their learning. Building upon the learning sciences literature, we can identify the best practices for collaborative activities in developing project-based work [90] as follows: (1) learning-appropriate goals, (2) scaffolds that support both student and teacher learning, (3) frequent opportunities for formative self-assessment and revision, and (4) social organizations that promote participation and result in the sense of agency. In this study, we helped students identify appropriate learning goals for a deep understanding

of formal iterative design practices, develop scaffolds for learning these goals through instructor modeling and coaching, teaching embedded into the project work, and contrasting cases. Specifically, we helped students recognize the value of scaffolds, resources, and social structures that encourage and support prototype development.

Second, we want to stress the importance of teaching creative design in the HCI classroom. Our experience was that it was helpful to create a familiar topic for students to work on, in our case, a task to support students in a learning situation. As both HCI designers and students themselves, students find it easier to stand in their own shoes to think of creative and innovative ideas since they are designers and subject experts at the same time. As shown in Table 3, the developed tangibles in our paper have satisfied not only the requirements for a math class but also shown good possibilities for other applications. For example, *stayFOCUSed*, a light projector, has been designed to engage students in group collaboration. However, we can also use it as a portable device to display information (e.g., text, image, and icon), similar to a smart lamp. Altogether, when guided through a careful design of the learning experience, we found that students' creativity led to a quality of results that we could even use to gain insights relevant to HCI research.

*6.5. Limitations*

Educational experts evaluated the prototypes only through the author's introduction and user scenario videos. Compared to physical touch and play with tangibles, our methods might produce some misunderstanding, even though the experts have a good knowledge of collaborative learning and understand the design scenarios in this study. The reason for inviting educational experts online was because of COVID-19. It was impossible to have physical contact with the prototypes (e.g., touch and play). Therefore, the participants must know the design purposes well.

The initial HCI design was primarily for showing the conceptual ideas and might sacrifice the actual sizes or materials. Therefore, the prototypes in this study have some evident defects since they are early prototypes. However, educational experts might not realize the relationship between a prototype fully. A polished product could lead to negative comments such as the "prototype is too bulky to be practical", and the "prototype is rough"; these prototypes are in their early states (not close to a product), and we need to evaluate and refine them further before conducting in-the-wild experiments. It would not be reasonable to invest in in-the-wild studies at this stage.

It was hard to convincingly demonstrate the practical effects of tangible design at the current stage. Tangible learning is a relatively new technology or concept; it benefits learning due to embodied [25] and playful [12] characteristics. However, to test its real effects on learning requires field studies. Explaining the effects from theoretical or lab-based perspectives is valuable, but it is still not convincing without evidence from practical and in-field studies. However, tangibles have a problem implementing large-scale user studies because the prototype development costs time and money and needs technical maintenance.

The current prototypical implementations serve as starting points to gather the presented feedback from the experts. We can now improve their implementations using the human-centered design process based on their feedback. This will allow us to improve the quality and robustness of them and finally study them in field tests. Only then will we be able to verify how well the tangibles help to orchestrate collaborative learning.

## 7. Conclusions

Tangibles represent an embodied approach for sharing information in the classroom. To design collaborative learning orchestration tangibles, we extracted the design space of tangibles for classroom orchestration of collaborative learning. The design space we extracted for tangible orchestration of collaborative learning contains 4 essential elements with 12 dimensions that designers should consider when designing tangibles for classroom orchestration of collaborative learning. We contributed to the design, implementation,

and evaluation of five tangibles for collaborative classroom activities with the design space in mind. We obtained valuable and insightful perspectives from ten educational experts by presenting five newly designed prototypes in interviews. The experts supported that tangibles were crucial for effective collaboration; a collaborative effort might not automatically emerge in a tangible collaborative environment. Further, we found it is crucial to create flexible, simple, and fun tangibles for orchestrating collaborative activities in an actual class. In summary, we conclude that educational experts support tangibles for collaboration in classrooms and see a promising future for them when designed right.

Future studies are needed to explore integrating serious and social fun into tangibles for orchestrating collaborative learning. Here, we proposed two challenges to designing tangibles: (1) tangible design is activity-determined but should be available for other use cases; (2) usability and affordance of tangibles are hard to predict in actual classes.

**Author Contributions:** Conceptualization, Y.L., B.R., T.W. and H.H.; methodology, Y.L. B.R., T.W. and H.H.; investigation, Y.L., B.R. and T.W.; writing—original draft preparation, B.R. and Y.L.; writing—review and editing, A.K., T.W., S.M. and H.H.; supervision, S.M. and H.H. All authors have read and agreed to the published version of the manuscript.

**Funding:** Authors are supported by the Elite Network of Bavaria (K-GS-2012-209).

**Institutional Review Board Statement:** Not applicable.

**Informed Consent Statement:** Not applicable.

**Data Availability Statement:** Not applicable.

**Acknowledgments:** Seventeen master students who took our *Practical Tangible Light* courses at LMU Munich developed the tangible tools in the paper. *stayFOCUSed*: Aleksa Ristic, Sarah Muser, Agnes Reda, and Tianyang Lu; *Group Hexagon*: Sybil Bast, Anna-Carina Gehlisch, Laura Haller, and Ina Klautke; *Tower*: Havy Ha, Mariam Ali Hussain, Jennifer Herner, and Vanessa Balzer; *Glowing Wand*: Anke Pellhammer, Jonathan Haudenschild, and Sandra Wackerl; *Remolight*: Meng Liang and Teodora Mitrevska.

**Conflicts of Interest:** The authors declare no conflict of interest.

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
