# Peer review of "Designing Tangible as an Orchestration Tool for Collaborative Activities"

_mti, doi:10.3390/mti6050030_

Round 1

Reviewer 1 Report

This paper presents a case of use of tangibles in educational setting, with an extensive review of literature and presentation and evaluation of five prototypes.

The prototypes are interesting and the presentation of the results of the expert assessement provides some hint for future developments. 

The experimental results are good, but they of course suffer from the limited size of users.

It might be useful to provide a technical description of the architecture behind the various prototypes, especially if they can all rely on some common infrastructure and mechanisms.

As for related work, the authors might consider the following two papers as relevant:

Patrick BaudischTorsten BeckerFrederik Rudeck:
Lumino: tangible blocks for tabletop computers based on glass fiber bundles. CHI 2010: 1165-1174

Federico GelsominiKamen KanevReneta P. BarnevaPaolo BottoniDonna Hurst TatsukiMaria Roccaforte:
BYOD Collaborative Storytelling in Tangible Technology-Enhanced Language Learning Settings. IMCL 2018: 22-33

Author Response

Reviewer 1

Comment 1:

It might be useful to provide a technical description of the architecture behind the various prototypes, especially if they can all rely on some common infrastructure and mechanisms.

Response 1: 

We have added the technical descriptions for each prototype.

Comment 2:

The authors might consider the following two papers

Response 2: 

Thanks for pointing out the missing works; we have added these two papers as part of our related work.

Comment 3:

The experimental results are good, but they of course suffer from the limited size of users.

Response 3: 

We agree with the concern, the current versions are in a prototypical stage of development. Thus, before putting them into the hands of users, we aimed for expert feedback to then improve the tangible. In the next step, we will further develop them and test them in the field. We added this as future work.

General Response:

We improved the paper’s readability and structure to deliver our contribution better.

Reviewer 2 Report

This paper presents design guidelines and prototypes of 5 tangible devices used for classroom orchestration. The prototypes are evaluated by 10 experts and their qualitative ratings are recorded. 

The paper is well written and easy to understand. The design section is interesting. But the limitations on evaluation, however, limit the significance of the conclusion. The videos might not be appropriate to completely evaluate the tangibles (as also acknowledged by the authors). 

The tangibles were never evaluated in a classroom settings with actual users so the reader doesn't know how the tangibles will fair from a student point of view. 

Author Response

Reviewer 2

Comment 1:

the limitations on evaluation, [...] The tangibles were never evaluated in a classroom setting with actual users so the reader doesn't know how the tangibles will fair from a student’s point of view. 

Response 1:

We agree with the assessment by the reviewer that it would be beneficial to test the prototypes in classrooms. Similar to Comment 3 by Reviewer 1, we argue that the prototypes in their early state (not close to a product) are not there yet, and need to be evaluated more closely before handing out for in-the-wild testing. In fact, Reviewer 3 points this out “it would not be reasonable to invest in it at this stage of the project.”  We added this to future work.

General Response:

We improved the paper’s readability and structure to deliver our contribution better.

Reviewer 3 Report

The paper reports on a study conducted with the aim of evaluating the efficacy of tangible interaction as an orchestration tool for collaborative learning.

The paper is very well written and well structured, and the study is conducted in a rigorous way. Research questions are clearly identified. Limitations are singled out and discussed. The discussion emphasizes in a clear way pro and cons detected by the experts.

Tha lack of user-based evaluation is a weak point, but, on the other hand, on the basis of the experts' review it would not be reasonable to invest in it at this stage of the project. 

As far as I'm concerned, I share the doubt of some of the experts about the real advantage and the added value of the presented tangible tools compared with a traditional interaction. 

Experts' outcome suggests the need of additional work, but the paper is anyhow interesting for the scientific community in its present form, also from a methodological point of view.

Author Response

Reviewer 3

Comment 1:

I share the doubt of some of the experts about the real advantage and the added value of the presented tangible tools compared with a traditional interaction. 

Response 1:

The revised manuscript clarified that this has to be investigated in future work.

General Response:

We improved the paper’s readability and structure to deliver our contribution better.

Round 2

Reviewer 2 Report

I thank the authors for their modifications. 

I would still suggest adding how video evaluation is feasible for tangible devices compared to real world evaluation. 

Author Response

We have added the explanations for it.